# Unveiling Diversity for Quality Traits in the Indian Landraces of Horsegram [*Macrotyloma uniflorum* (Lam.) Verdc.]

**DOI:** 10.3390/plants12223803

**Published:** 2023-11-08

**Authors:** Manju Kumari, Siddhant Ranjan Padhi, Sushil Kumar Chourey, Vishal Kondal, Swapnil S. Thakare, Ankita Negi, Veena Gupta, Mamta Arya, Jeshima Khan Yasin, Rakesh Singh, Chellapilla Bharadwaj, Atul Kumar, Kailash Chandra Bhatt, Rakesh Bhardwaj, Jai Chand Rana, Tanay Joshi, Amritbir Riar

**Affiliations:** 1The Graduate School, ICAR—Indian Agricultural Research Institute, PUSA, New Delhi 110012, India; manjukhakhil@gmail.com (M.K.); siddhant.padhi1@gmail.com (S.R.P.); 2ICAR—National Bureau of Plant Genetic Resource, PUSA, New Delhi 110012, India; sushilchourey76@gmail.com (S.K.C.); vishalkondal4@gmail.com (V.K.); veena.gupta@icar.gov.in (V.G.); mamta.arya@icar.gov.in (M.A.); yasin.jeshima@icar.gov.in (J.K.Y.); rakesh.singh2@icar.gov.in (R.S.); 3ICAR—Indian Agricultural Research Institute, New Delhi 110012, India; swapnilbiochem24@gmail.com (S.S.T.); chbharadwaj@yahoo.co.in (C.B.); atul.kumar@icar.gov.in (A.K.); 4ICAR—Indian Agricultural Statistics Research Institute, New Delhi 110012, India; negiankita2508@gmail.com; 5The Alliance of Bioversity International & CIAT—India Office, New Delhi 110012, India; j.rana@cgiar.org; 6Department of International Cooperation, Research Institute of Organic Agriculture FiBL, 5070 Frick, Switzerland; tanay.joshi@fibl.org (T.J.); amritbir.riar@fibl.org (A.R.)

**Keywords:** underutilized legume, nutritional profiling, variability, multivariate analysis, PCA, HCA

## Abstract

Horsegram (*Macrotyloma uniflorum* [Lam.] Verdc.) is an underutilized pulse crop primarily cultivated in South Asian countries like India, Nepal, and Sri Lanka. It offers various nutraceutical properties and demonstrates remarkable resilience to both biotic and abiotic stresses. As a result, it has emerged as a promising crop for ensuring future food and nutritional security. The purpose of this study was to assess the nutritional profile of 139 horsegram germplasm lines obtained from 16 Indian states that were conserved at the National Gene Bank of India. Standard analytical methods, including those provided by the Association of Official Analytical Chemists (AOAC), were used for this investigation. The study revealed substantial variability in essential nutrients, such as protein (ranging from 21.8 to 26.7 g/100 g), starch (ranging from 26.2 to 33.0 g/100 g), total soluble sugars (TSSs) (ranging from 0.86 to 12.1 g/100 g), phenolics (ranging from 3.38 to 11.3 mg gallic acid equivalents (GAEs)/g), and phytic acid content (ranging from 1.07 to 21.2 mg/g). Noteworthy correlations were observed, including a strong positive correlation between sugars and phenols (r = 0.70) and a moderate negative correlation between protein and starch (r = −0.61) among the studied germplasm lines. Principal component analysis (PCA) highlighted that the first three principal components contributed to 88.32% of the total variability, with TSSs, phytates, and phenols emerging as the most significant contributors. The cluster analysis grouped the accessions into five clusters, with cluster III containing the accessions with the most desirable traits. The differential distribution of the accessions from north India into clusters I and III suggested a potential geographical influence on the adaptation and selection of genes. This study identified a panel of promising accessions exhibiting multiple desirable traits. These specific accessions could significantly aid quality breeding programs or be directly released as cultivars if they perform well agronomically.

## 1. Introduction

Proteins play a crucial role in repairing and building body tissues across all organisms. They serve as key components of enzymes and hormones, regulating vital metabolic processes necessary for the normal growth and development of the human body [1]. Recent studies indicate that plant-derived proteins may be more beneficial in reducing the risks of cardiovascular diseases compared with their animal-derived counterparts [2,3]. As awareness of health, environmental concerns, and compassion for animals have grown, there has been a significant increase in the global vegetarian population. For vegetarians, legumes, including horsegram, stand out as some of the most affordable and widely available sources of proteins.

India leads in global pulse production, contributing 25.67 million tons (29.57% share) of the total world production of 86.83 million tons, surpassing China, with 4.8 million tons (5.53% share), and Canada, with 4.33 million tons (4.98% share) production [4]. The data also reveal that chickpeas (11.91 million tons), beans (6.12 million tons), red gram (4.32 million tons), lentils (1.49 million tons), and peas (0.88 million tons) are the most commonly grown pulses in India, comprising over 96% of the total pulses produced in the country.

In India, pulses are typically grown under rainfed conditions in marginal soils, as irrigated fertile zones are primarily dedicated to rice and wheat cultivation. Only 23.10% of the total area under pulses in India is irrigated, compared with 56.96% for cereals. Consequently, the productivity of pulses has increased only 2-fold since 1950, in contrast with a 4.6-fold increase in the productivity of cereals [5]. To enhance the productivity of pulses, it is essential to overcome the yield barrier through genetic enhancement using elite germplasm lines and by promoting the cultivation of underutilized pulse crops through scientific research. This approach will unlock the full potential of pulses to contribute to sustainable food production and improve the livelihoods of smallholder farmers.

Recognizing the nutritional and food security potential of pulses, the United Nations designated 2016 as the “International Year of Pulses” to raise public awareness about the nutritional and food security potential of pulses.

Horsegram (*Macrotyloma uniflorum* [Lam.] Verdc.), an indigenous legume native to India, is an underutilized crop that has played a vital role in subsistence farming in India [6]. Thriving in diverse ecological conditions, horsegram has the potential for wider cultivation and utilization in sustainable agriculture. Despite its adaptability, horsegram has been less studied compared with other major food crops. Encouraging research and promoting its utilization could unlock its full potential as a nutritious, environmentally resilient crop contributing to food security and sustainable agricultural practices.

It is crucial to note that horsegram is also grown as forage and green manure in Australia, parts of Africa, and the West Indies on lands with marginal and poor soils. In India, the states of Karnataka (34%), Tamil Nadu (18%), Maharashtra (18%), Andhra Pradesh (16%), and Odisha (16%) account for 90 to 95 percent of the area under horsegram, either as a sole crop or a mixed crop with millets. Horsegram is consumed in various forms in India, including boiled or fried, as flour mixed with other cereals, or as a split pulse [7]. Recent innovations have explored the utilization of horsegram in various food products, showcasing its potential for enhancing nutritional value. For example, germinated and roasted horsegram flour mixed with cereal flours has been tested for the preparation of chapattis, breads, cookies, and biscuits, demonstrating high antioxidant activities and sensory qualities. Additionally, a novel fungal fermented food product named “kaulath” has been developed from horsegram seeds and recommended as a potential ingredient in formulated food products [8]. The extrusion prepared using horsegram and lentil flour has been suggested to create ready-to-eat (RTE) desserts with high protein, digestibility, and viscosity [9]. Furthermore, a protein concentrate with enhanced quality and digestibility has been formulated for supplementary diets [10].

In terms of nutritional content, horsegram is notable for its higher carbohydrate content (57.24 g/100 g) with high starch content (47.96 g/100 g), along with substantial quantities of minerals and vitamins, similar to other pulses [11]. However, it contains certain antinutritional components, including phytic acid (10.2 mg/g), polyphenols (14.3 mg GAE/g), and oligosaccharides (26.8 mg/g) [12]. Importantly, horsegram exhibits resilience against abiotic stress factors such as drought, salinity, and heavy metal stress, showcasing its ability to thrive across a broad spectrum of temperatures—an advantage often elusive to other crops [13]. This resilience positions horsegram as an optimal choice for rainfed agriculture, a significant segment in India comprising around 52% of its net sown area and contributing to around 40% of the nation’s total food supply [14]. Despite its potential, this underutilized legume has yet to receive comprehensive research in the context of its nutraceutical properties to supplement the food system as a source of nutraceuticals or functional foods. Hence, the present study was designed to systematically assess nutritionally for the selected 139 distinct germplasm accessions representing 16 Indian states categorized as landraces and conserved at the National Gene Bank of India, located at ICAR-NBPGR New Delhi, India.

## 2. Results

### 2.1. Nutritional Composition

In this study, we conducted a comprehensive nutritional composition analysis of 139 horsegram accessions based on dry-weight measurements. Our statistical analysis revealed substantial variability in the nutritional composition among the accessions. Notably, the crop exhibited a high protein content of 23.7 g/100 g, alongside relatively lower soluble sugars and significant phenolic compounds (Table 1). For a detailed breakdown of the nutritional profile of each accession, specific observations can be found in Appendix A. We further assessed the variability using standard deviation (SD) and coefficient of variation (CV) metrics. For protein content, the SD and CV (0.99 and 4.16%) indicated that the data clustered closely around the mean, showcasing minimal variation in protein content. In contrast, soluble sugars exhibited a higher level of variability, with SD and CV of 2.50 and 44.55%, respectively, compared with protein, starch, and phenols. The most notable variability was observed for phytic acid (SD: 4.36, CV: 42.92%), suggesting the potential for selecting the best accessions based on this trait. To provide additional context, we compared our results with the Indian Food Composition Table 2017 (IFCT2017). Our findings indicate that the mean content of protein, total sugars, phytic acid, and phenols is high, while starch content is relatively low in the germplasm compared with the IFCT2017. This analysis underscores the importance of understanding and utilizing the observed variability in horsegram nutritional composition, which can inform strategies for breeding and cultivation aimed at enhancing the nutritional value of this valuable pulse crop.

### 2.2. Correlation Analysis

The Pearson’s correlation coefficient (r) was calculated to understand the relationships between different traits in the horsegram germplasm collection. The results are presented in Table 2 and can be visually explored in the correlogram (Appendix A). A high positive correlation (r = 0.70, *p* < 0.01) was observed between the total soluble sugars and phenols. This strong positive relationship suggests that as the total soluble sugars increase, there is a corresponding increase in phenolic compounds. On the other hand, the total protein content showed a negative correlation with starch (r = −0.61, *p* < 0.01), total soluble sugars (r = −0.33, *p* < 0.01), and phenols (r = −0.26, *p* < 0.01). These negative correlations, though varying in strength, suggest that higher protein content is associated with lower levels of starch, soluble sugars, and phenols. Interestingly, there was almost no correlation between proteins and phytic acid (r = 0.14, *p* < 0.09). This indicates that protein content does not significantly affect the levels of phytic acid in horsegram. Understanding these correlations is essential for tailoring breeding and cultivation strategies to optimize the nutritional composition of horsegram, aiming to enhance its value as a food crop.

### 2.3. Principal Component Analysis 

The first three components from principal component analysis, each possessing eigenvalues surpassing 0.999, collectively accounted for a substantial 88.32% of the overall variability. This distribution is supported by a scree plot (Appendix A). In a more detailed breakdown, the principal components individually explained 39.73%, 28.59%, and 19.99% of the overall variance, respectively, as detailed in Appendix A. It is worth highlighting that PC-1 exhibited the most notable factor loadings, with prominent contributions from TSSs (−0.78), phenols (−0.70), and starch (−0.62). These factors collectively underpinned 34.96%, 28.02%, and 21.87%, respectively, of the variance encapsulated by PC-1, as elucidated in Appendix A. Similarly, PC-2’s main factor loadings were from protein (0.62) and phytate (−0.61), explaining 36.03% and 35.10% of the variability accounted for by PC-2 (Appendix A), respectively. Remarkably, phytate alone emerged as the most crucial character, with a factor loading of 0.78, explaining 63.44% of the variability depicted by the third principal component. To further visualize the relationships between the principal components, we provide a PCA plot (Figure 1). These plots offer additional insights into the grouping of accessions.

### 2.4. Hierarchical Clustering Analysis

The agglomerative HCA, using Ward’s method, grouped the accessions into five clusters (Figure 2). Table 3 shows the mean values of the five biochemical traits across the five clusters generated using cluster analysis.

## 3. Discussion

### 3.1. Nutritional Analysis

#### 3.1.1. Total Protein Content

Studies on horsegram germplasm diversity have been limited. Previous research has shown a range of protein content in horsegram spanning from 21.7 to 24.9% [7,11,12]. However, these studies were primarily conducted on market samples. In contrast, our analysis is based on 139 germplasm lines (landraces) representing various distinct collecting sites in India. The median protein content observed in our study (23.8%) closely aligns with the mean value (23.7%). This allowed us to identify 70 accessions with protein content surpassing the mean value of 23.7%, showcasing potential for utilization in breeding high-protein horsegram cultivars.

#### 3.1.2. Total Starch Content

Starches, sugars, and dietary fibers constitute the primary dietary carbohydrates. Legumes serve as secondary sources of carbohydrates, constituting about 50–60% of their dry weight [7]. Starch content in legumes, though essential, has received less focus than protein content and antinutritional elements. Starch takes precedence as the most abundant available carbohydrate in legumes (22–45%), followed by dietary fibers (4.3–25%) and oligosaccharides (1.8–18%) [15,16]. In our investigation, we found the starch content among horsegram accessions (26.20–33.00 g/100 g) to be below previously reported values, which is preferable as legumes are primarily consumed as a source of protein rather than carbohydrates. Additionally, starch content shows a negative correlation with protein content in legumes [17,18]; thus, accessions with lower starch content are expected to contain more proteins, aligning with comparison with the IFCT2017 values for horsegram [11].

#### 3.1.3. Total Soluble Sugar Content

Total soluble sugars in legumes encompass monosaccharides, disaccharides, and oligosaccharides, including raffinose family oligosaccharides (RFOs) [19]. RFOs, though considered antinutritional factors causing flatulence, are widespread in legumes and other plant parts. Nevertheless, RFOs play a constructive role by stimulating beneficial bacteria proliferation while suppressing harmful pathogens. Furthermore, they exhibit anti-allergic, antidiabetic, and anti-obesity properties [20]. In horsegram and other legumes, oligosaccharides constitute a significant portion of the total soluble sugars [7]. Thus, the accessions with high total soluble sugars are likely to have elevated levels of RFOs. The variation in total soluble sugars (0.86 to 12.10 g/100 g) is attributed to unique genotypic makeups influenced by adaptive natural selection processes in specific environmental niches. RFOs in plants regulate seed germination, vigor, and longevity and confer resistance to various biotic and abiotic stresses, including desiccation tolerance [20,21]. Germplasm accessions with high soluble sugars may be screened against various stresses to develop stress-tolerant cultivars.

#### 3.1.4. Total Phenol Content

Phenolics, a diverse group of secondary metabolites abundantly found in plants, primarily reside in the seed coat of legumes [22]. These phenolic compounds play a crucial role in determining the color, taste, and flavor of foods [7]. Initially deemed antinutritional, phenolics are now recognized for their beneficial attributes such as antioxidant, anti-aging, anti-inflammatory, and antiproliferative properties. These qualities reduce the risks of diabetes, cancer, cardiovascular diseases, and skin disorders by managing oxidative stress [23,24]. In the context of horsegram, the overall phenolic content ranges from 3.49 mg GAE/g [7] to 14.3 mg GAE/g [12]. Altitudinal variations in phenolic content have also been observed [25]. The phenolic content variability observed in the current investigation may be attributed to the wide range of seed coat colors among the accessions. Our findings align with earlier studies indicating that cultivars of garden pea [26] and common bean [27] with darker seed coats exhibit higher concentrations of phenolics.

#### 3.1.5. Total Phytic Acid Content

Phytic acid (PA) is the primary form of phosphorus storage in plants, constituting up to 80% of the total stored seed phosphorus in cereals, legumes, oilseeds, and nuts [28]. Despite its significant nutritional role, PA is often labeled as an antinutritional factor due to its capacity to form complexes with minerals, proteins, and starch, reducing their bioavailability [29]. For monogastric creatures like humans and chickens, there exists a deficiency of enzymes capable of effectively breaking down phytate, leading to incomplete metabolism and subsequent excretion of PA in their fecal matter [28]. Notably, PA has also been implicated in the emergence of the “hard to cook” phenomenon often witnessed in pulses [29]. However, the mineral-binding capacity of PA has been associated with reducing the risk of certain cancers, promoting heart health, and managing kidney stones. Additionally, PA finds application in preserving the green color of vegetables, preventing lipid peroxidation, and minimizing enzymatic browning in fruits and vegetables [30]. Understanding and managing phytic acid levels in legumes can significantly impact their nutritional value and overall food quality.

Phytic acid is considered an important antinutritional compound in pulses; however, cereals such as wheat also show a remarkable amount of phytic acid (0.40–2.10%) [31]. Chickpea, the chief pulse crop of India, contains phytic acid ranging from 0.009 to 4.06 mg/g [32]. Other legume species, such as blackgram, pigeonpea, lentil, mungbean, and cowpea, demonstrate phytic acid concentrations ranging from 5.74 to 19.0 mg/g [18,33,34,35,36]. The widest range of phytic acid (3.85 to 49.5 mg/g) has been reported for a core set of 203 common bean germplasm accessions collected from diverse areas of Jammu and Kashmir in India [37]. The PA content in horsegram has been reported to be 10.20 mg/g [12]. In contrast, the horsegram accessions analyzed in our study displayed levels ranging from 1.07 mg/g (BSP 15-1) to 21.2 mg/g (IC071733). Furthermore, approximately 34 accessions from the present study were identified to contain less PA than the first quartile value of 6.87 mg/g, which is comparable to, and in some cases less than, the reported values for other pulses. This shows that the germplasm lines evaluated in the present investigation may be useful for developing low-phytate cultivars. Furthermore, food processing methods, including fermentation and the utilization of phytase enzymes, have been proposed as strategies to diminish the levels of phytic acid in food products [28]. Simple methods like dehulling, cooking, germination, and roasting have also been proposed to reduce both phytic acid and phenolics in legume seeds [7,38]. Understanding and managing phytic acid levels in legumes can have significant implications for enhancing their nutritional value and overall food quality.

### 3.2. Multivariate Analysis and Identification of Promising Germplasms

In our multivariate analysis, we observed a notable negative correlation between seed protein and starch content consistent with previous studies on various crops, e.g., on triticale [17], soybeans [39,40], field peas [41], and cowpeas [18]. This correlation may be explained by the influence of glucose-6-phosphate/phosphate translocator (GPT) levels, which affect starch synthesis and assimilate partitioning into storage proteins [42]. Notably, phytic acid did not show a significant correlation with other desirable traits, suggesting the possibility of independently selecting genotypes with low phytic acid and high protein content. The positive association between sugars and phenolics has been attributed to their shared synthesis pathways [43]. The PCA effectively compressed the data and highlighted that sugars, phenols, and starch may warrant more emphasis in diversity studies. 

Cluster analysis revealed distinctive traits across various clusters. Significantly, cluster III presented a substantial mean protein content (23.7 g/100 g) coupled with a moderate level of phenolics (6.65 mg GAE/g), sugars (4.96 g/100 g), and starch (28.6 g/100 g). Furthermore, it exhibited the lowest mean phytic acid (4.86 mg/g). This combination positions cluster III as a promising candidate for the development of horsegram cultivars with desirable attributes. In contrast, cluster V exhibited the highest mean phytic acid (16.5 mg/g), trailed by cluster IV (14.8 mg/g), rendering them less suitable for the cultivation of low-phytate variants. Cluster I (43 accessions) and cluster III (33 accessions) formed a significant portion (about 55%) of the total 139 accessions. Notably, these clusters included accessions from Uttarakhand (23/29), Himachal Pradesh (10/12), Chhattisgarh (8/16), Andhra Pradesh (8/13), and Madhya Pradesh (6/14). The majority of accessions of unknown origin were grouped in cluster II (Appendix A). The remaining accessions in clusters IV and V did not display any specific geographical trends, possibly due to their smaller sample sizes. The analysis further revealed that the accessions from the Indian states located at higher altitudes, i.e., Uttarakhand, Arunachal Pradesh, and Himachal Pradesh, contained elevated levels of sugars and phenolics but below-average levels of protein (Appendix A). These results align with the previous reports [25] and support the role of sugars and phenolics in stress management at higher altitudes [20,21]. The accessions from Kerala, West Bengal, Chhattisgarh, and Andhra Pradesh were found to be protein rich. The exotic accession from Ethiopia contained higher levels of proteins, phenols, and phytic acid, coupled with a low sugar content, and can be used in breeding programs.

By employing multivariate analysis, promising accessions with superior agronomic traits were identified (Table 4). Selected horsegram genotypes exhibited high levels of protein, starch, and sugars coupled with low levels of phytic acid and polyphenols, making them excellent candidates for further breeding programs. Each trait and their combinations yielded five superior accessions (Table 4). For instance, IC023482 had the highest protein content (26.7 g/100 g), lowest phenols (3.38 mg GAE/g), low sugars (1.31 g/100 g), moderate starch (29.0 g/100 g), and high phytic acid (14.2 mg/g), presenting an opportunity to develop ready-to-eat (RTE) foods with roasted pulses catering to diabetic consumers. While accessions IC022795 and IC139512 shared similar nutritional profiles with higher protein, moderate starch, and low sugars and phenols, they differed in phytic acid content. On the other hand, accession IC019431 exhibited high protein (24.2 g/100 g) and low phytic acid (2.83 mg/g), phenols (4.98 mg GAE/g), and sugars (3.64 g/100 g) with moderate starch (30.2 g/100 g). The cultivar BSP 15-1 showcased the lowest phytic acid (1.07 mg/g), and moderate starch (28.4 g/100 g) and protein (23.9 g/100 g), but higher levels of sugars (9.53 g/100 g) and polyphenols (9.04 mg GAE/g). By leveraging these germplasm accessions through multivariate approaches, the development of diverse horsegram cultivars has the potential for catering to different consumer needs.

## 4. Materials and Methods

### 4.1. Plant Material

A total of 139 germplasm accessions of horsegram (*Macrotyloma uniflorum* [Lam.] Verdc.) were sourced from the Indian National Gene Bank at ICAR-NBPGR, New Delhi, India (Appendix A). Seeds were planted at NBPGR Regional Station, Bhowali, Uttarakhand, India (29.550 N, 79.600 S) during Kharif-2018, following recommended agricultural practices. Selected accessions like BSP 15-1, CRHG-19, CRIDA-18R, VL-15, and VL-19 were used as reference checks, and an exotic collection from Ethiopia (EC501564) was included to verify geostratification. Sowing of the germplasm lines was performed in the first week of July 2018; seeds were harvested at physiological maturity during October and November 2018, and then stored in the short-term storage facility at Bhowali. The biochemical analysis and subsequent data analyses were conducted at the NBPGR headquarters in New Delhi, India. Dried seeds were homogenized using a Foss cyclone mill equipped with a 1 mm sieve, and this homogenized flour was used for all biochemical analyses.

### 4.2. Methodology

#### 4.2.1. Estimation of Total Proteins

The DUMAS method (AOAC 992.23) [44] was employed to estimate the total N_2_ concentration using the Elementar Rapid MAX N exceed nitrogen auto analyzer. A 100 mg sample was weighed in a stainless steel crucible using a Sartorius Electronic weighing balance (Model ATX224R, Göttingen, Germany). The samples underwent combustion at 950 °C in a high-oxygen environment. Following combustion, nitrogen oxides were converted to nitrogen using a copper–platinum catalyst and estimated with a thermal conductivity detector. The instrument was calibrated using HPLC-grade L-aspartic acid (Sigma A9256-100G, Sigma-Aldrich, Saint Louis, MO, USA) for nitrogen recovery. The nitrogen concentration was then converted to a protein percentage using a Jones’ factor of 6.25.

#### 4.2.2. Estimation of Total Soluble Sugars, Starch, Phenols, and Phytic Acid

Total soluble sugars, total starch, total phenols, and phytic acid were estimated using standard and official methods with minor modifications as applied to cowpea germplasm [18] as and described here briefly. Total soluble sugar was estimated in an 80% ethanol extract obtained from a 0.1 g sample using the anthrone reagent method [45]. Total starch content was estimated in the residue of the sample obtained after the extraction of total soluble sugar using the Megazyme total starch assay kit in accordance with AOAC 996.11 [44]. Total phenolic content was estimated in an 80% ethanol extract obtained from a 0.1 g sample using the Folin–Ciocalteu reagent (FCR) method [46]. Phytic acid was estimated using the K-PHYT standard assay method (Megazyme International Ireland 2019) (AOAC 986.11) [44]. Total and free phosphorus content were estimated separately using an ascorbic acid–molybdate color reagent.

#### 4.2.3. Statistical Analysis

Multivariate techniques were used to analyze the biochemical variations and classify the 139 horsegram accessions based on five quantitative traits, employing the software package “R-Studio” [47]. This included calculating Pearson’s correlation coefficients (r) and conducting principal component analysis using default algorithms. Additionally, hierarchical cluster analysis using Ward’s method was performed to organize the dataset into distinct groups.

## 5. Conclusions

Legumes are vital components of the human diet, providing essential proteins, minerals, vitamins, and bioactive elements. The global production of seed legumes has witnessed a significant increase, and their versatility in the food processing industry has led to the creation of ready-to-eat (RTE) foods and health-promoting products. With superior nutritional compositions compared with commonly grown pulses such as chickpeas, redgram, greengram, and blackgram, horsegram has substantial potential to be promoted as a primary pulse crop in India. Based on these findings, we conclude that horsegram holds great promise as a future crop worth exploring. Among the selected accessions, IC023482 exhibited the highest protein content (26.70 g/100 g) and the lowest phenols (3.38 mg GAE/g), while accessions IC022795, IC139512, IC019431, and BSP 15-1 demonstrated diverse nutritional profiles. The exceptional horsegram genotypes identified in this study, featuring high protein, starch, and sugars, and low phytic acid and polyphenols, are excellent candidates for breeding programs and the development of diabetic-friendly ready-to-eat (RTE) foods.

## Figures and Tables

**Figure 1 plants-12-03803-f001:**
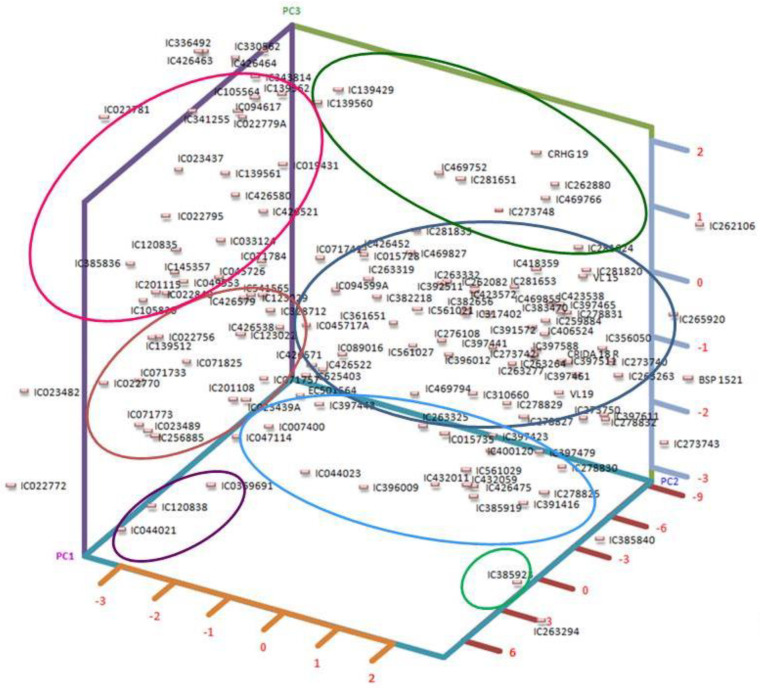
Three-dimensional PCA plot grouping accessions based on first three PCs.

**Figure 2 plants-12-03803-f002:**
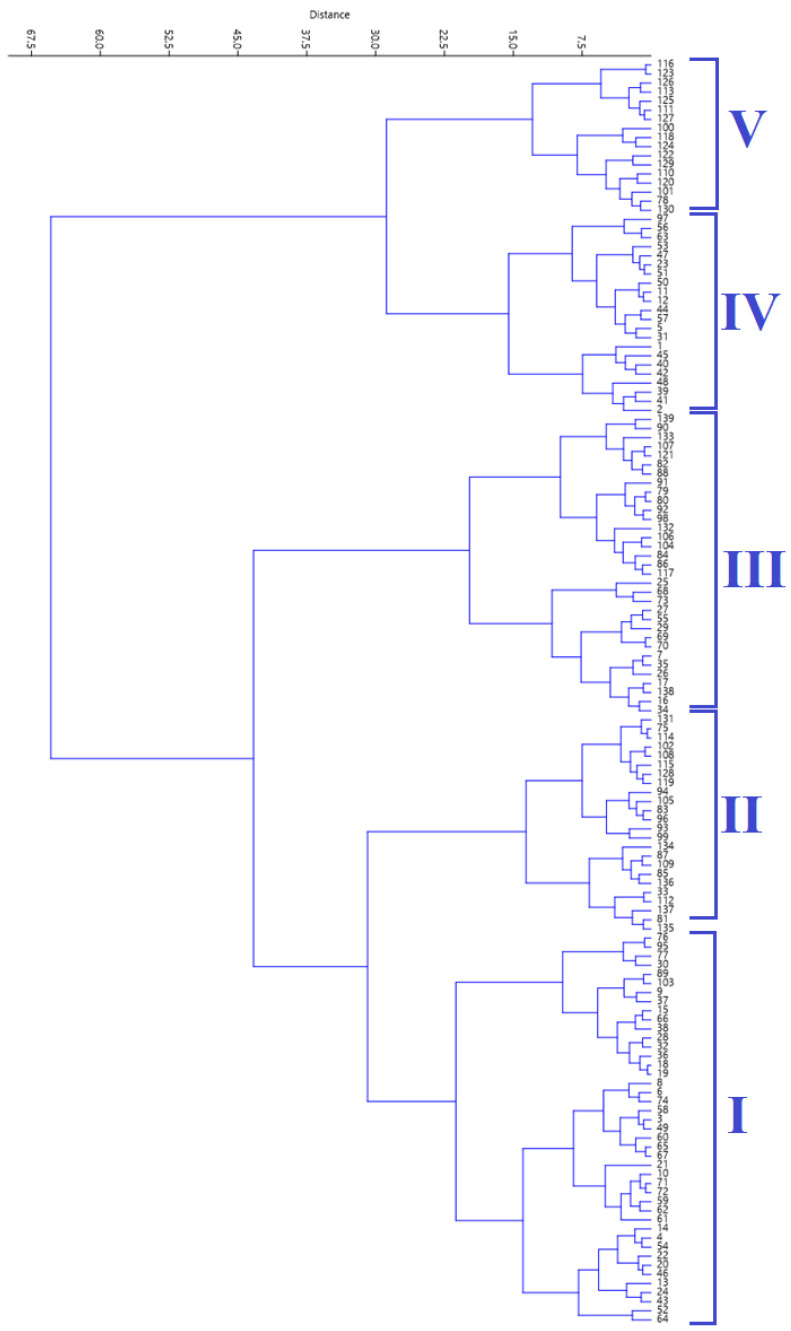
Cluster diagram generated using hierarchical cluster analysis.

**Table 1 plants-12-03803-t001:** Measures of variability for 139 horsegram accessions.

Estimate of Variability	Protein (g/100 g)	Starch (g/100 g)	Total Soluble Sugars (g/100 g)	Phytic Acid (mg/g)	Phenols (mg GAE/g)
Median	23.8	29.9	5.47	10.2	7.01
Mean	23.7	29.8	5.61	10.2	7.00
Standard deviation	0.99	1.36	2.50	4.36	1.71
Coefficient of variation %	4.16	4.56	44.6	42.9	24.5
Minimum value	21.8	26.2	0.86	1.07	3.38
Accession with minimum value	IC139429	IC0369691	IC022772	BSP 15-1	IC023482
Maximum value	26.7	33.00	12.10	21.2	11.3
Accession with maximum value	IC023482	IC426463	IC265920	IC071733	IC278825
IFCT 2017	21.7	47.9	1.99 *	3.39	3.32

* sum of total sugars and oligosaccharides.

**Table 2 plants-12-03803-t002:** Pearson’s correlation coefficients matrix.

Trait	Protein	Starch	Sugars	Phytic Acid	Phenols
Protein	1.00				
Starch	−0.61 **	1.00			
Sugars	−0.33 **	0.02	1.00		
Phytic Acid	0.14	0.07	−0.05	1.00	
Phenols	−0.26 **	−0.08	0.70 **	−0.10	1.00

** Correlation is significant at the 0.01 level.

**Table 3 plants-12-03803-t003:** Cluster mean values of the hierarchical cluster analysis using Ward’s method.

Properties	HCA Cluster Mean					Overall Mean
	I	II	III	IV	V	
Number of accessions	43	24	33	22	17	
Protein (g/100 g)	23.4	24.0	23.7	23.3	24.7	23.7
Starch (g/100 g)	29.9	29.6	29.8	30.5	29.3	29.8
Sugar (g/100 g)	7.15	3.50	4.96	8.02	2.9	5.61
Phytic acid (mg/g)	9.36	10.1	4.86	14.8	16.5	10.2
Phenol mg (GAE/g)	8.17	5.65	6.65	8.32	4.91	7.00

**Table 4 plants-12-03803-t004:** Trait-wise superior germplasms identified from the study.

SN	Trait/s	Accessions Identified as Superior
1	Low phytic acid (mg/g)	BSP 15-1 (1.07), IC262106 (1.66), IC281653 (1.88), IC019431 (2.83), IC281820 (3.41)
2	Low phenols (mg GAE/g)	IC023482 (3.38), IC022781 (3.76), IC022795 (3.81), IC105564 (3.85), IC139512 (3.93)
3	High sugars (g/100 g)	IC265920 (12.1), IC273743 (10.7), IC397511 (10.2), IC262106 (9.64), BSP 15-1 (9.53)
4	High protein (g/100 g)	IC023482 (26.70), IC256885 (26.2), IC044021 (26.1), IC139512 (26), IC022795 (25.8)
5	High starch (g/100 g)	IC426463 (33.00), IC273748 (33.0), IC343814 (32.6), IC469766 (32.4), IC418359 (32.2)
6	Moderate protein, low phenols, low sugars, high phytic acid	IC022781 (23.4 g/100 g, 3.76 mg GAE/g, 1.60 g/100 g, 10.1 mg/g)
7	High protein, low phenols, low sugars, high phytic acid	IC023482 (26.70 g/100 g, 3.38 mg GAE/g, 1.31 g/100 g, 14.2 mg/g)
8	High protein, low phenols, low sugars, moderate phytic acid	IC022795 (25.8 g/100 g, 3.81 mg GAE/g, 2.46 g/100 g, 5.21 mg/g)
9	High protein, low phenols, low sugars, low phytic acid	IC139512 (26.0 g/100 g, 3.93 mg GAE/g, 3.60 g/100 g, 3.93 mg/g)
		IC019431 (24.2 g/100 g, 4.98 mg GAE/g, 3.64 g/100 g, 2.83 mg/g

## Data Availability

All data published in this paper.

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
