# Peer review of "Unveiling Diversity for Quality Traits in the Indian Landraces of Horsegram [Macrotyloma uniflorum (Lam.) Verdc.]"

_plants, 2023, doi:10.3390/plants12223803_

Round 1
Reviewer 1 Report
Comments and Suggestions for Authors
This study covers the chemical composition of horsegram, however, it is required a clear setup of the rationale of study and experimental design such as specific reasons for the different site collection and what expectation is from other cultivars regarding the correlation between protein and starch. Here are my specific comments:
L2 Please confirm if this format is correct.
L3 The title is a bit ambiguous. Try to revise it based on case study or specific results.
L39 Provide full name
L41 Again, provide full name
L42 Regarding what? correlations found between substances after abiotic stresses or site factors?
L56 required references as well.
L58 Provide some references regarding this.
L62 Related to reference [1]?
L65 Please provide numeric value of cultivation %.
L69 Reference
L169 What is implication of hierarchical clustering in this research?
L178 I do not see any differences between the results and the discussion part in this manuscript. Please discuss with many references, case studies and your opinion by combining them.
L182 Generally higher contents than other cultivars or horsegram from other regions?
L202 Please discuss about it more? why and what is implication regarding this?
L216 RFOs
L220 Still wondering if you found any correlation among genotype, collection site factors and environmental factors such as temperature, rainfall, drought, soil type and so on.
L265 Even if it is better than discussion 3.1., it is likely to be required to write more concisely and provide what was significant findings and what was main implication from component analysis and hierarchical clustering analysis, respectively.
L266 Due to which reasons? Please cite this paper or something similar and improve it in this discussion part.
https://ift.onlinelibrary.wiley.com/doi/full/10.1111/1541-4337.13141
L320 When harvested and how long was it cultivated?
L327 What does it mean for? Association of Official Agricultural Chemists?
L358 Any documentation or previous studies that horse gram is more promising for food resources rather than animal feedstock these days?

Comments on the Quality of English LanguageThere's no significant issue in English writing.
Author Response
Kindly find the attached file.

Reviewer 2 Report
Comments and Suggestions for Authors
The authors assessed the nutritional profile of 139 horsegram germplasm lines obtained from 16 Indian states, using standard and AOAC methods. This is useful to facilitate quality breeding program or can be directly released as cultivars, if performed better agronomically. It is well oragnized through the whole manuscript. It is better if the authors could supplement the roles of the native climatic and geographic factors of the horsegram germplasm lines in affecting the nurtritional traits.
Author Response
Kindly find the attached file.

Round 2
Reviewer 1 Report
Comments and Suggestions for Authors
The authors provided enough answers with a revised version of the manuscript. I believe this manuscript is satisfied with my peer review undergoing. Thank you for your endeavor.
Comments on the Quality of English LanguageN/A